# Emerging Role of Protein O-GlcNAcylation in Liver Metabolism: Implications for Diabetes and NAFLD

**DOI:** 10.3390/ijms24032142

**Published:** 2023-01-21

**Authors:** Ziyan Xie, Ting Xie, Jieying Liu, Qian Zhang, Xinhua Xiao

**Affiliations:** 1Department of Endocrinology, Key Laboratory of Endocrinology, Ministry of Health, Peking Union Medical College Hospital, Chinese Academy of Medical Sciences & Peking Union Medical College, Beijing 100730, China; 2Department of Medical Research Center, Peking Union Medical College Hospital, Chinese Academy of Medical Sciences & Peking Union Medical College, Beijing 100730, China

**Keywords:** O-GlcNAc, post-translational modification, liver metabolism, insulin resistance, diabetes, NAFLD

## Abstract

O-linked b-N-acetyl-glucosaminylation (O-GlcNAcylation) is one of the most common post-translational modifications of proteins, and is established by modifying the serine or threonine residues of nuclear, cytoplasmic, and mitochondrial proteins. O-GlcNAc signaling is considered a critical nutrient sensor, and affects numerous proteins involved in cellular metabolic processes. O-GlcNAcylation modulates protein functions in different patterns, including protein stabilization, enzymatic activity, transcriptional activity, and protein interactions. Disrupted O-GlcNAcylation is associated with an abnormal metabolic state, and may result in metabolic disorders. As the liver is the center of nutrient metabolism, this review provides a brief description of the features of the O-GlcNAc signaling pathway, and summarizes the regulatory functions and underlying molecular mechanisms of O-GlcNAcylation in liver metabolism. Finally, this review highlights the role of O-GlcNAcylation in liver-associated diseases, such as diabetes and nonalcoholic fatty liver disease (NAFLD). We hope this review not only benefits the understanding of O-GlcNAc biology, but also provides new insights for treatments against liver-associated metabolic disorders.

## 1. Introduction

The liver is the largest internal organ, and acts as an important central organ for various physiological processes. The liver is responsible for blood storage and allocation, as well as blood factor synthesis, detoxification, immune defense, whole-body metabolism, and energy homeostasis [1].

The processing and metabolism of nutrients provides energy and supports numerous biological processes. The liver is a critical site for carbohydrate, lipid, and protein metabolism [1]. Glucose is taken up and stored as glycogen in the liver. After high blood sugar consumption, glycogenolysis is triggered, exporting glucose into the blood circulation, which maintains the stability of the blood glucose concentration [2]. The liver also carries out the synthesis, oxidization, packaging, and secretion of lipids and lipoproteins. Fatty acid oxidization is a major source of energy, but the liver also provides energy from the process of ketogenesis [3]. Furthermore, the amino acids absorbed by the digestive tract undergo protein synthesis, deamination, and transamination in the liver; the synthesized protein enters the blood circulation, while nitrogenous waste is disposed in the form of urea [4]. Finally, hepatocytes produce bile acid and secrete bile. The liver is also the main hub for vitamin and mineral storage [1].

Given the unique and complex role of the liver, the disruption (e.g., lifestyle, alcohol, virus, hereditary factor, etc.) of liver metabolism and functions leads to several metabolic disorders, such as nonalcoholic fatty liver disease (NAFLD), which has presented a high prevalence in recent years in China [5], and has attracted growing concerns. Epidemiological data indicate that NAFLD and diabetes often coexist [6], and comorbidities result in severe metabolic consequences (nonalcoholic steatohepatitis, cirrhosis, and liver failure) and an increased risk of cardiovascular complications [7,8,9,10].

O-linked b-N-acetyl-glucosaminylation (O-GlcNAcylation) is a type of protein glycosylation. It was first reported in the 1980s, broadening the knowledge of protein modification by sugars. O-GlcNAcylation is characterized as a process of adding an O-linked b-N-acetylglucosamine (O-GlcNAc) molecule to the threonine or serine residues of nuclear, cytoplasmic, and mitochondrial proteins under enzymatic modulation [11]. The enzymes mediating O-GlcNAcylation consist of O-GlcNAc transferase (OGT) and O-GlcNAcase (OGA), which add and remove O-GlcNAc from target proteins, respectively [12]. Similar to phosphorylation, O-GlcNAcylation is a widespread and dynamic post-translational modification that has been discovered to contribute to multiple cellular activities, including cell survival, cell proliferation, transcription, epigenetic remodeling, metabolism regulation, mitochondrial function, circadian modulation, and several intercellular signaling pathways [13]. In addition, O-GlcNAcylation is a nutrient-driven process due to its close link with the hexosamine biosynthesis pathway (HBP). The HBP shares the first two steps with glycolysis, and is a sub-branch of glucose metabolism. The end-product of the HBP, uridine diphosphate N-acetylglucosamine (UDP-GlcNAc), is the only donor substrate for O-GlcNAcylation, which affects the level of O-GlcNAcylation [14]. HBP integrates many metabolic pathways, and the UDP-GlcNAc and O-GlcNAcylation levels are influenced by the flux of glucose, amino acids, lipids, and nucleotides [15].

Recent studies have connected the roles of aberrant O-GlcNAcylation, O-GlcNAc cycling enzymes, and the HBP to the pathogenesis and progression of liver-associated disorders, including hepatocellular carcinoma (HCC) [16,17,18], fatty liver diseases [19,20,21], diabetes [22,23], and diabetic complications [24,25]. This review discusses the role of O-GlcNAcylation in liver metabolism, and highlights the implications of O-GlcNAcylation in liver metabolic diseases, including diabetes, nonalcoholic fatty liver disease (NAFLD), and associated liver fibrosis, thus providing evidence for pathogenesis and potential therapeutic targets.

## 2. Overview of O-GlcNAcylation

### 2.1. Hexosamine Biosynthesis Pathway (HBP)

The HBP is one of the glucose metabolic pathways by which the body uses glucose, glutamine, acetyl-CoA, aspartic acid, and UTP to synthesize the final product, namely, UDP-GlcNAc, the glycosyl donor for O-GlcNAcylation [26] (Figure 1). It has been reported that 2–5% of glucose flux enters the HBP in adipocytes [14]. UDP-GlcNAc incorporates various metabolic pathways, and is modulated partially by the levels of substrates, thus exerting effects on O-GlcNAcylation [27]. Oxaloacetate and aspartate supplementation increase the levels of UDP-GlcNAc and induce O-GlcNAcylation in hepatoma cells [16]. The level of glutamine in adipocytes has been shown to influence glycolysis and the HBP, which builds a link between glutamine and O-GlcNAcylation [28]. The rate-limiting step of the HBP is catalyzed by L-glutamine:D-fructose-6-phosphate aminotransferase (GFAT), converting fructose 6-phosphate into glucosamine 6-phosphate [29]. GFAT is the primary target of the HBP, regulating the influx of glucose. There are two isoforms of GFAT in humans, namely, GFAT1 and GFAT2, and each isoform has significantly distinct tissue expression and functions [30]. The activity of GFAT largely relies on the level of its substrates, namely, fructose-6-phosphate and glutamine. The glucosamine-6-phosphate and UDP-GlcNAc products exhibit negative feedback on GFAT. At the transcriptional level, several transcription factors have been shown to regulate GFAT, such as X-box binding protein-1 (XBP-1) [31] and specificity protein 1 (Sp1) [32]. At the post-translational level, GFAT has various phosphorylation sites. AMP-dependent protein kinase (PKA) has been reported to modulate the phosphorylation of GFAT1 and GFAT2 on Ser205 [33] and Ser202 [34], respectively. AMPK induces the phosphorylation of GFAT1 at Ser-243, and thus suppresses its enzymatic activity [35].

### 2.2. O-GlcNAc Cycling Enzymes: OGT and OGA

In addition to the HBP, the levels of O-GlcNAcylation are also dependent on OGT and OGA, which are the only cycling enzymes identified to catalyze this dynamic and reversible process. OGT adds O-GlcNAc to proteins using UDP-GlcNAc as the sugar donor, and this activity is highly correlated with the level of UDP-GlcNAc [36,37]. The gene encoding OGT is conserved and localized on the X chromosome in humans. There are three isoforms of OGT in mammals, as follows: nucleocytoplasmic OGT (ncOGT), mitochondrial OGT (mOGT), and short OGT (sOGT). The three isoforms differ in subcellular distributions and structures [38]. OGT contains a tetratricopeptide repeat (TPR) domain at the N-terminus that is distinct among the isoforms, resulting in substrate specificity and dimerization. The catalytic domain is located at the C-terminus, which catalyzes the O-GlcNAcylation of target proteins [39,40]. OGT is extensively expressed in multiple tissues and cell types. Under normal conditions, the expression of OGT is not high in the liver, while overexpressed OGT leads to pathological changes and hepatic disorders. Increased OGT in the liver has been reported to inhibit the expression of insulin signaling genes, such as *IRS1* and *Akt*, contributing to insulin resistance [41]. Furthermore, elevated OGT and O-GlcNAcylation are suggested as hallmarks of HCC [42]. Conversely, decreased OGT is also linked with liver disorders. Patients with alcoholic liver cirrhosis have reduced OGT and O-GlcNAc levels [43]. Liver-specific OGT knockout mice exhibit hepatocyte necroptosis, and develop liver fibrosis at an early age [43].

OGA is responsible for the removal of O-GlcNAc, and is encoded by the *MGEA5* gene on chromosome 10 [44]. The OGA protein contains an N-acetyl-b-D-glucosaminidase region in the N-terminus and a pseudohistone acetyltransferase (HAT) region in the C-terminus [45]. Alternative splicing generates a variant of OGA, called short OGA (sOGA), that lacks the HAT domain [46]. sOGA has significantly weaker hexosaminidase activity, and is located in the nucleus and on the surface of lipid droplets [47,48]. Similar to OGT, OGA is also widely distributed in a variety of tissues. OGT and OGA maintain an ‘optimal’ range of O-GlcNAcylation, ensuring a rapid response to stimuli [49]. Consequently, aberrant expression and function of OGT and OGA leads to changes in O-GlcNAcylation beyond this range, resulting in hepatocyte dysfunction and increased susceptibility to liver disorders [20].

### 2.3. O-GlcNAc-Modified Proteins

O-GlcNAcylation occurs in nuclear, cytoplasmic, and mitochondrial proteins [13]. It has been widely reported that O-GlcNAc modification affects multiple target protein functions [13,50,51]. The key pattern by which O-GlcNAcylation modulates protein function is described herein. First, O-GlcNAcylation stabilizes proteins by blocking ubiquitination and degradation or influencing protein folding and aggregation. O-GlcNAcylation enhances the stability of the p53 protein, resulting in the accumulation and activation of p53 [52,53]. In addition, O-GlcNAcylation plays a thermoprotective role for Sp1 by suppressing the heat-related misfolding of the Sp1 protein [54]. Second, O-GlcNAcylation impacts enzyme activity. O-GlcNAc modification of glucose-6-phosphate dehydrogenase (G6PD), which catalyzes the rate-limiting step of the pentose phosphate pathway, augments G6PD activity and increases glucose flux [55]. Similarly, pyruvate dehydrogenase (PDH) activity also has a positive correlation with the level of O-GlcNAcylation [56]. Decreased O-GlcNAcylation results in reduced PDH activity and low pyruvate utilization in mitochondria. Third, O-GlcNAcylation influences the transactivation and transrepression ability of transcription regulators. The liver X receptor (LXR) modulates the transcription of its downstream lipogenic gene, carbohydrate response element-binding protein (*ChREBP*). The O-GlcNAc-modified LXR confers significantly increased transactivation of the *ChREBP* promoter, and OGT knockdown suppresses LXR-mediated transactivation [57]. Additionally, O-GlcNAcylation at S662 of period circadian clock 2 (PER2) blocks phosphorylation in this region and enhances PER2 suppressor activity [58]. Fourth, protein–protein interactions are also indispensable targets of O-GlcNAcylation. O-GlcNAc modification induces the nuclear translocation of p27, a tumor suppressor, thus disrupting the interaction between Cyclin/CDK and p27, which promotes cell proliferation in hepatocellular carcinoma [59]. Yes-associated protein 1 (YAP) is a transcriptional coactivator of the Hippo signaling pathway, and is reported to be O-GlcNAcylated at serine 109. O-GlcNAcylation disrupts binding to the upstream kinase LATS1, thus improving the transcriptional activity of YAP and facilitating tumorigenesis [60].

## 3. O-GlcNAcylation in Liver Metabolism

O-GlcNAcylation has been reported to regulate metabolism in several organs. In this section, we summarize the metabolic pathways influenced by O-GlcNAcylation in the liver, and discuss the molecular mechanisms (Figure 2).

### 3.1. Glucose Metabolism

The liver is a major organ involved in balancing glucose hemostasis. Accumulating evidence shows that O-GlcNAcylation plays a role as a nutritional sensor to modulate hepatic glucose metabolism processes.

Glucokinase (GCK) acts as the initial gatekeeper for the liver glucose-sensing system, catalyzing glucose to glucose-6-phosphate (G6P), which is subsequently utilized in glycolysis and glycogen synthesis [61]. In diabetic mouse livers, elevated O-GlcNAcylation has been found to increase the level of GCK protein [62]. Mechanistically, the addition of O-GlcNAc stabilizes the GCK protein, thus enhancing GCK expression and activity. Type 2 diabetes is correlated with the impairment of glucose utilization by the liver, and partially inactivated GCK contributes to maturity onset diabetes of the young type 2 (MODY2) [63]. Therefore, GCK expression is pivotal in maintaining glucose utilization and hemostasis. O-GlcNAcylation provides a new regulatory mechanism for GCK in the liver.

O-GlcNAc modification also alters the expression and activity of key proteins involved in liver gluconeogenesis and glycogen synthesis. Carbohydrate response element-binding protein (ChREBP) mediates hepatic glucose and lipid metabolism by binding to a ChoRE sequence of a target gene. O-GlcNAcylation has been reported to modulate ChREBP activity and binding affinity for target genes [64,65]. Under high glucose conditions, O-GlcNAcylation of forkhead box protein O1 (FoxO1), a transcription factor involved in various metabolic pathways, induces the transcription of glucose 6-phosphatase (G6Pase) in the liver, leading to increased hepatic glucose production and aggravated glucotoxicity [66,67]. PGC-1α is a major regulatory factor in gluconeogenesis. OGT-mediated O-GlcNAcylation prevents the degradation of the PGC-1α protein by stimulating its interaction with deubiquitinase, and ultimately enhances gluconeogenesis [68]. Conversely, SIRT1, a NAD+-dependent deacetylase, regulates the expression of multiple downstream genes, including FOXO1 and PGC-1α. Inhibition of the O-GlcNAc-modified form of SIRT1 contributes to hyperglycemia by promoting the transcription of *PGC-1α* and *FOXO1*, resulting in the increased expression of phosphoenolpyruvate carboxykinase (PEPCK) and G6Pase, therefore enhancing liver gluconeogenesis [69]. In addition, estrogen-related receptor γ (ERRγ), also a master positive regulator of liver gluconeogenesis, has been suggested to be O-GlcNAcylated in the starved state. This modification facilitates the stabilization of the ERRγ protein, which recruits ERRγ to gluconeogenic gene promoters [70]. Similarly, p53 has been reported to be stabilized by O-GlcNAc modification in the liver during starvation. p53 induces the transcriptional activation of PCK1, thus enhancing hepatic glucose synthesis [53].

In addition to liver glucose production, recent findings have also indicated that O-GlcNAcylation exerts an effect on hepatic glycogen metabolism [71,72,73]. Elevated O-GlcNAcylation of glycogen synthase is found in HepG2 cells upon glucose deprivation. Increased O-GlcNAc modification reduces glycogen synthase (GS) activity by 60% to adapt to the low-glucose environment and mobilize glucose utilization [72]. Studies have also indicated that O-GlcNAcylation impacts the expression and activity of glycogen synthase kinase-3β (GSK3β) by indirectly modulating its phosphorylation [73,74]. GSK-3β is a negative regulator of hepatic glycogen metabolism. GSK-3β inhibits the activity of glycogen synthase by phosphorylation, thus reducing the synthesis of hepatic glucose and elevating the blood glucose concentration [75]. Under acute cold stress, increased O-GlcNAcylation activates AKT in the liver, an upstream signaling pathway that regulates various proteins associated with glucose homeostasis, which then promotes the phosphorylation and activation of GSK-β [71]. 6-Phosphofructo-2-kinase/fructose-2,6-biphosphatase 2 (PFKFB2), a key enzyme in glycolysis, is likewise activated by O-GlcNAcylated AKT, leading to increased glucose use in cold stress [71,76]. Moreover, glycogenolytic enzymes have also been suggested to be regulated by O-GlcNAc modification. Liver glycogen phosphorylase (PYGL) requires O-GlcNAcylation to function. Glucose and insulin block the O-GlcNAcylation of PYGL, contributing to inhibited glycogen breakdown, while glucagon has the opposite effect [77]. Taken together, these data reveal a wide role for O-GlcNAcylation in hepatic glucose metabolism, which affects glucose sensing and induces glucose production and utilization, as well as abolishes glycogen synthesis.

### 3.2. Lipid Metabolism

Most lipids contain, or are derived from, fatty acids, while other lipids, such as phospholipids and cholesterol, are crucial constituents of membranes. Triglycerides act as the main form for the storage and transport of fatty acids in the liver. The fatty acids synthesized from de novo lipogenesis by hepatocytes account for 30% of fatty acids [78]. O-GlcNAcylation participates in the regulation of liver lipogenesis. In vivo and in vitro studies have indicated that under high-glucose stimulation, suppressing OGT and GFAT inhibits lipid accumulation, while an inhibitor of OGA promotes lipid deposition in both HepG2 cells and animal liver [79]. Pei et al. also confirmed that the HBP and O-GlcNAcylation affect lipogenesis. Knockdown of GFAT1, the rate-limiting enzyme of the HBP, significantly decreases the production of saturated fatty acids, unsaturated fatty acids, and cholesterol, as measured by liquid chromatography–mass spectrometry (LC–MS) [80]. FAS is the determinant enzyme driving de novo lipogenesis (DNL). The correlation between liver FAS and O-GlcNAc modification has been observed in diabetic mouse models. O-GlcNAcylation regulates FAS expression and activity at both the transcriptional and protein levels. OGA inhibitors enhance the O-GlcNAc modification of FAS, leading to increased RNA levels and blocked ubiquitination degradation of the FAS protein [81]. Another study has demonstrated that OGT and FAS are colocalized in the cytoplasm and interact physically and functionally [21]. In addition, knocking down or blocking GFAT-1, which decreases UDP-GlcNAc, the substrate for O-GlcNAcylation, also reduces FAS expression and activity [82]. In addition to FAS, acetyl-CoA carboxylase (ACC) is the rate-limiting enzyme of de novo lipogenesis, which converts acetyl-CoA into malonyl-CoA. A high-fat diet has been suggested to facilitate O-GlcNAc modification, thus hindering the phosphorylation of ACC, resulting in activated ACC and enhanced lipid synthesis in the liver [83].

O-GlcNAcylation affects de novo lipogenesis in the liver not only by regulating lipid synthesis enzymes, such as FAS and ACC, but also by the upstream transcription factors controlling these enzymes, such as ChREBP, sterol regulatory element-binding protein (SREBP-1), and liver X receptor (LXR). The O-GlcNAc modification of ChREBP in response to hyperglycemia has been previously discussed in liver glucose metabolism [64]. Increasing O-GlcNAc-modified ChREBP by overexpressing OGT leads to enhanced protein stability and transcription of the *FAS*, *ACC*, and *SCD1* lipogenic genes. Conversely, overexpression of OGA, contributing to reduced ChREBP O-GlcNAcylation, inhibits the expression of FAS and ACC [84]. Host cell factor 1 (HCF-1) has been identified as a ChREBP regulatory protein. HCF-1 is O-GlcNAcylated in the liver upon glucose stimulation, which binds with ChREBP, induces the recruitment of OGT, and promotes ChREBP O-GlcNAcylation and transactivation of lipogenesis genes [85]. SREBP1 has also been predicted to modulate multiple lipogenic genes [86]. OGT and O-GlcNAcylation have been reported to regulate SREBP-1, thus affecting its transcriptional targets and lipid metabolism in many organs [87]. Moreover, under insulin treatment, O-GlcNAcylation of specificity protein 1 (Sp1) facilitates binding to the promoter of SREBP1, which activates the SREBP1/ACC1 pathway, leading to lipid droplet deposition [88]. Liver X receptor (LXR) also plays an important role in liver de novo lipogenesis. O-GlcNAc modification improves the ability of LXR to transactivate SREBP-1c, therefore enhancing the expression of target lipogenic genes and synthesis of triacylglycerol [57]. Intriguingly, LXR can be O-GlcNAc-modified and control the O-GlcNAcylation of other proteins by interacting with nuclear OGT. Animal model experiments have shown that knockout of LXR decreases hyperglycemia-induced nuclear O-GlcNAcylation and inhibits the activity of hepatic ChREBP and lipogenic gene expression [89]. In addition to transcription, O-GlcNAcylation also influences mRNA splicing of lipogenic genes by modifying serine/arginine protein-specific kinase 2 (SRPK2). O-GlcNAc-SRPK2 promotes efficient splicing of target genes, thus increasing the level of lipogenic enzymes and production of fatty acids [80].

Furthermore, O-GlcNAc modification may also exert effects on cholesterol metabolism through interaction with LXR. Enhanced hepatic LXRα O-GlcNAcylation has been detected both in a cell model and in a streptozotocin-induced refed diabetic mouse model [57,89,90]. LXRα plays a vital role in cholesterol homeostasis. Loss of LXRα induces significant cholesterol accumulation in the liver and accelerates atherogenesis in mice fed a high-fat diet [91]. LXRα modulates the expression and activity of cholesterol 7α-hydroxylase (Cyp7a1), the rate-limiting enzyme in the bile acid synthesis pathway, and induces the transformation of cholesterol into bile acid [92]. In addition, LXRα also regulates the expression of key factors involved in cholesterol excretion (G5 and G8 ATP-binding cassette transporters) and reverse cholesterol transport (ABCA1 and Apo E) [93]. O-GlcNAcylation of LXRα is a novel and vital mechanism used to regulate LXR-dependent gene expression, indicating the important role of LXRα in lipid hemostasis.

### 3.3. Bile Acid Metabolism

Bile acid synthesis and secretion constitute dispensable cholesterol consumption in the liver [94]. Bile acids are important components of bile, and are mainly found in the enterohepatic circulatory system. After a meal, bile acids are secreted in the intestine from the gallbladder, and then either excreted in feces or reabsorbed by the intestine and enter enterohepatic circulation, maintaining a cycling bile acid pool. Farnesoid X receptor (FXR), which belongs to the nuclear receptor family, represents the key regulator of genes involved in bile acid production and transportation in the liver and the gut [95]. In vivo and in vitro studies have demonstrated that high glucose stimulation enhances O-GlcNAc modification of FXR, resulting in improved protein stability, transcriptional function, and chromatin binding ability of FXR [64]. O-GlcNAcylation of FXR in the fed state significantly affects the expression of FXR target genes involved in bile acid metabolism, which upregulates the genes responsible for bile acid secretion and elimination (*BSEP*, *KNG1*, *Ostβ*, and *Mdr1*) as well as suppresses the expression of genes associated with bile acid synthesis (*Cyp7a1* and *Cyp8b1*), thus leading to a decrease in liver bile acid content [96]. In summary, O-GlcNAcylation plays a crucial role in modulating the functions of nuclear receptors, such as FXR, which in turn influences the target genes associated with hepatic and intestinal bile acid metabolism.

### 3.4. Urea Metabolism

Ureagenesis is an important detoxification process in the liver that converts the deleterious products of protein breakdown of ammonia into urea. Carbamoyl phosphate synthetase 1 (CPS1) is the rate-limiting enzyme of ureagenesis, which catalyzes the direct incorporation of ammonia into urea cycle intermediates in mitochondria [97]. A recent study has reported that the O-GlcNAcylation level of CPS1 is elevated in hyperammonemia [98]. Increased O-GlcNAcylation on specific threonine residues enhances the catalytic activity of CPS1, thus promoting ureagenesis during hyperammonemia. Blocking OGA reduces systemic ammonia in mouse models of liver diseases. Interestingly, another study has also found that CPS1 is one of the most O-GlcNAcylated proteins in the livers of aged mice. However, O-GlcNAc modification of the serine 537A site of CPS1 suppresses its enzymatic activity, and silencing of OGT recovers CPS1 activity [99]. These two studies indicate opposite results; however, O-GlcNAcylation regulates a broad range of protein functions, and the same protein with different modification sites may have various effects.

## 4. O-GlcNAcylation in Diabetes

Diabetes mellitus is a complex metabolic disorder mainly characterized by hyperglycemia. The liver plays a critical role in blood glucose homeostasis by regulating glycogenesis, glycogenolysis, gluconeogenesis, and ketogenesis [1]. Insulin and other hormones modulate these events by insulin signaling and gene expression, leading to inhibition or stimulation of glucose production [100]. Accumulating evidence has shown that O-GlcNAcylation alters insulin signaling, hepatic glycogen metabolism, and gluconeogenesis (Figure 3).

Enzymes catalyzing O-GlcNAc modification are associated with insulin resistance. The levels of OGT and GFAT are significantly elevated in the livers of patients with type 2 diabetes and positively correlated with blood glucose and HOMA-IR [53]. Additionally, in livers of STZ-treated diabetic rats, the level of OGT is altered and is restored by the addition of insulin, indicating the potential role of OGT in insulin signaling [101]. Overexpression of OGT in the liver influences the phosphorylation of insulin-responsive proteins, namely, Akt and insulin receptor substrate 1 (IRS-1), which inhibits the kinase activity of Akt and induces the serine phosphorylation of IRS1, thus resulting in insulin resistance and dyslipidemia [41]. In addition, conditional knockout of OGA contributes to disrupted insulin sensitivity, glucose tolerance, and hyperleptinemia [73]. Similarly, transgenic mice overexpressing GFAT in the liver also exhibit glucose intolerance and insulin resistance [102,103]. Moreover, proteins involved in insulin signaling have also been found to be O-GlcNAcylated. High-fat intervention increases O-GlcNAcylation of protein tyrosine phosphatase 1B (PTP1B) in HepG2 cells, which then enhances the phosphorylation of insulin receptor and IRS1, resulting in impaired insulin sensitivity, while blocking PTP1B O-GlcNAcylation by site mutation induces hepatic glucose uptake and improves insulin resistance [104]. SIRT1 is also necessary in insulin signaling, and its loss of function is linked with insulin resistance. Suppressed glycosylation of hepatic SIRT1 leads to systemic insulin resistance, hyperglycemia, and hepatic inflammation, contributing to liver dysfunction and metabolic disorders, including diabetes [69].

O-GlcNAcylation has also been reported to alter hepatic glucose metabolism in diabetes. Excessive hepatic gluconeogenesis remains a major cause of hyperglycemia in diabetes. Evidence has shown that OGT and the HCF-1 complex cooperatively induce gluconeogenesis by upregulating the O-GlcNAcylation and stability of PGC-1α [68]. Liver-specific deletion of OGT and HCF-1 improves glucose homeostasis in diabetic mice. In addition, OGT also triggers hepatic gluconeogenesis by O-GlcNAc modification of CRTC2, a transcriptional coactivator of CREB, resulting in enhanced activity of CREB on gluconeogenic genes [105]. Blocking the O-GlcNAcylation of CRTC2 inhibits the positive effects of glucose on gluconeogenesis. Similarly, O-GlcNAcylated FOXO1 is increased in the liver under hyperglycemic conditions, leading to activated transcriptional activity, which upregulates the expression of gluconeogenic genes, such as *G6P*, and leads to excessive hepatic glucose production [66,67]. The O-GlcNAcylation of p53 is also upregulated in the liver of diabetic patients, which transcriptionally activates PCK1, thus inducing hepatic glucose synthesis and output [53]. Furthermore, O-GlcNAcylation influences glycogen synthase both in normoglycemia and diabetes. O-GlcNAcylated glycogen synthase becomes resistant to insulin stimulation, which reduces hepatic glycogen synthesis and increases circulating glucose levels in diabetic mice [106].

In addition, it is widely believed that inflammation and oxidative stress are involved in the pathogenesis and development of liver dysfunction in diabetes. Liver-produced retinol-binding protein 4 (RBP4) is increased and associated with inflammation in multiple metabolic diseases [107]. A recent study has indicated that O-GlcNAcylation is involved in RBP4 overproduction and inflammation in the livers of diabetic and obese mice as well as high-glucose-cultured hepatocytes [108]. O-GlcNAcylation of retinol-binding protein receptor 2 (RBPR2) is elevated in the liver under hyperglycemic conditions, leading to decreased RBP4 binding activity and a disrupted cellular retinol cascade, thus triggering inflammation in the liver. Deletion of OGT recovers the disruption of the retinol cascade and inflammation induced by high glucose in hepatocytes. Moreover, elevated O-GlcNAcylation of antioxidant enzymes, including superoxide dismutase (SOD) and catalases (CAT), has been detected in the livers of streptozotocin (STZ)-induced diabetic animals, leading to exaggerated reactive oxygen species (ROS) and activation of the p65 inflammatory transcription factor [109]. In addition, lowering the level of O-GlcNAcylation recovers the activity and expression of antioxidant enzymes, which alleviates liver oxidative stress in diabetes. Another study has also reported that liver-specific deletion of OGT recovers glutathione (GSH) replenishment and biosynthesis as well as prevents liver injury [110].

Several studies have made progress in the clinical translation of O-GlcNAcylation in diabetes [111,112,113,114]. Differences in gene expression involved in O-GlcNAcylation and the HBP, including *OGT*, *OGA*, *GFPT1*, and *GFPT2*, have been detected between diabetic and nondiabetic individuals [115], which may provide candidate susceptibility genes and diagnostic value for diabetes. Among the above genes, the *OGA* gene has been demonstrated to be a susceptibility locus for diabetes in the Mexican American population, as *OGA* mutations have been linked with an increased incidence of diabetes [111,112]. Furthermore, O-GlcNAcylation has shown potential in diagnostic efficacy. Elevated leukocyte O-GlcNAcylation has been reported to facilitate earlier detection of type 2 diabetes, and the level of O-GlcNAcylation in granulocytes distinguishes pre- and overt diabetes [113]. Additionally, in healthy young adults, O-GlcNAcylation exhibits a positive association with HOMA-IR, and has emerged as a more sensitive biomarker for insulin resistance than HbA1c [114]. Therefore, O-GlcNAc is a promising tool for predicting future metabolic status in healthy populations and earlier detection and diagnosis of diabetes.

Taken together, these studies indicate that O-GlcNAcylation exerts important effects on the development and progression of diabetes, suggesting that it may act as a potential biomarker for the prediction and early diagnosis of diabetes. However, further validation is still needed in a larger population.

## 5. O-GlcNAcylation in Nonalcoholic Fatty Liver Disease (NAFLD)

NAFLD remains the most common liver disorder in Western countries, and has a strong link with obesity, insulin resistance, and type 2 diabetes (T2D). The earliest feature of NAFLD is lipid accumulation (steatosis) within the cytoplasm of hepatocytes. As the disorder advances, a series of pathologic changes appear, including hepatocyte ballooning degeneration, inflammatory infiltration, and fibrosis, which is termed nonalcoholic steatohepatitis (NASH), and can ultimately progress to cirrhosis and hepatocellular carcinoma. Multiple studies have assessed the contribution of O-GlcNAcylation to the etiology of liver steatosis and fibrosis (Figure 4).

The imbalance between the synthesis and removal of fatty acids contributes to liver steatosis. DNL is the major pathway of lipid synthesis in the liver. Several enzymes and translational regulators involved in DNL have been reported to be influenced by O-GlcNAcylation and the HBP [80]. FAS and ACC are the rate-limiting enzymes in the DNL process. The expression and activity of liver FAS are directly and indirectly regulated by O-GlcNAcylation in ob/ob mice and in mice fed a high-carbohydrate diet [81]. O-GlcNAc modification affects FAS expression by directly protecting FAS from ubiquitin degradation and indirectly controlling transcription factors governing FAS transcription, including ChREBP, SREBP, and LXR, therefore enhancing FAS activity and hepatic lipid deposition under high-fat conditions. In addition, high fat may activate O-GlcNAcylation to interfere with the AMPK/ACC pathway and promote excessive lipid accumulation [83]. Elevated O-GlcNAcylation blocks the phosphorylation of AMPK and ACC, leading to activated ACC and increased lipogenesis both in livers under a high-fat diet and in hepatocytes incubated with fatty acids. Furthermore, O-GlcNAcylation of lipogenic regulators also contributes to altered hepatic lipid synthesis and fatty liver disorders. Host cell factor 1 (HCF-1) is enriched in the livers of NASH patients. Mechanistically, O-GlcNAcylation increases the expression of HCF-1, and O-GlcNAcylated HCF-1 then upregulates ChREBP, thus promoting DNL in hepatic steatosis [85]. Similarly, insulin deprivation induces Sp1 O-GlcNAcylation, which then transcriptionally activates SREBP1, resulting in excessive lipogenesis and lipid droplet formation in the liver [88]. LRH-1 also plays an important role in NAFLD via regulating fatty acid synthesis and oxidation, VLDL secretion, and liver inflammation [116]. In addition, LXR regulates de novo lipogenesis and cholesterol metabolism through the activation of SREBP-1c, and is thus involved in NAFLD progression [117]. Evidence has indicated that LXR is a target for O-GlcNAc modification under glucose and lipid metabolism disorders, and induction of O-GlcNAcylation of LXR is concomitant with upregulated expression of lipogenic genes [90]. Furthermore, Park et al. suggested that an OGT inhibitor reduces lipid accumulation induced by high glucose, while blocking OGA enhances lipid accumulation in HepG2 cells and the liver of zebrafish larvae, supporting the promising role of enzymes involved in O-GlcNAcylation and the HBP as targets for NAFLD therapy [79]. Overall, these findings corroborate the significance of O-GlcNAcylation in liver lipid synthesis and accumulation.

In addition, inflammation and oxidative stress are characteristic features during NAFLD progression [118]. With the deposition of lipids, lipotoxicity triggers ROS accumulation, inflammation, and related endoplasmic reticulum (ER) stresses in the liver [119]. Kwon et al. found that the global levels of O-GlcNAcylation and HBP flux proteins, including OGT and GFAT, are upregulated in the livers of methionine- and choline-deficient diet-induced NASH model mice. The highly upregulated O-GlcNAcylation activates the NF-κB pathway, resulting in an enhanced NF-κB-dependent inflammatory response, while blocking OGT inhibits the nuclear translocation of NF-κB p65 and exerts anti-inflammatory effects in NASH [120]. In addition, Kwon et al. also investigated the role of O-GlcNAcylation in oxidative stress using both in vitro and in vivo experiments; they reported that the levels of SIRT1 and SOD1 (antioxidant enzymes) are inversely correlated with high O-GlcNAcylation under high-fat conditions, while blocking OGT or GFAT by curcumin, which decreases the level of O-GlcNAcylation, recovers the expression of SIRT1 and SOD1, leading to reduced ROS [19]. Moreover, Sage et al. consistently observed that overexpression of GFAT exaggerates the ER stress response and downstream effects, including activation of lipid and inflammatory pathways, in hyperglycemia-induced hepatic steatosis [121]. A similar result has also been reported in NAFLD-related hepatocellular carcinoma with upregulated OGT, O-GlcNAcylation-induced NF-κB cascades, and activated endoplasmic reticulum stress [122]. However, hepatocellular carcinoma is beyond the scope of this review. Therefore, the anti-inflammation and antioxidant responses of O-GlcNAcylation pathway inhibition may shed light on the development of therapeutic strategies to alleviate NASH.

Exacerbation of NAFLD manifests as liver fibrosis, which is a result of extracellular matrix (ECM) accumulation, characterized by fibrous scar formation and hepatic architecture damage. Hepatic stellate cells (HSCs) are a type of fibrogenic cell, and the activation of HSCs is the major source of ECM in the liver and the key to liver fibrosis [123]. An in vitro study has demonstrated that O-GlcNAcylation is linked to HSC activation and collagen deposition. Inhibition of O-GlcNAcylation suppresses the proliferation and activation of HSCs by decreasing the expression of α-smooth muscle actin, collagen I, and collagen III [124]. Paradoxically, recent research has found that O-GlcNAcylation is reduced in HSCs during liver fibrosis. OGT knockdown, which decreases O-GlcNAcylated-SRF, an upstream translational regulator, results in increased levels of α-SMA contamination with fibrogenic genes, including acta2, Col1a1, and Col3A, by enhancing their promotor activity, indicating that O-GlcNAcylation is a negative regulator of HSC activation [125]. Altogether, both the activation and disruption of O-GlcNAcylation have been suggested to induce HSC activation. The former study used benzyl-α-GalNAc, a global O-GlcNAcylation inhibitor, and did not manipulate the level of O-GlcNAcylation [124]. The second study used OGT and OGA inhibitors as well as site mutations to block the O-GlcNAcylation of specific proteins [125]. Both studies used the same LX2 cell model. Therefore, the discrepancies may be due to different inhibitors (global and partial) and methods used to manipulate O-GlcNAc signaling (the first study did not quantify) as O-GlcNAcylation is a highly dynamic PTM, and acute or chronic changes in O-GlcNAc signaling may lead to different outcomes. Further study is required to address the role of O-GlcNAcylation in liver fibrosis.

Collectively, these findings indicate that O-GlcNAcylation is a potential target for the prevention and treatment of NAFLD, even though its exact contribution to liver fibrosis remains to be further explored.

## 6. Conclusions and Future Perspectives

Since the initial discovery of O-GlcNAcylation in 1984 [126], many efforts have been made to uncover the functions and roles of this post-translational modification. Studies have provided new insights into the importance of protein O-GlcNAcylation in metabolic control. In this review, we discussed the functions and regulatory mechanism of O-GlcNAcylation in liver metabolism, and focused on its role in the pathogenesis of diabetes and NAFLD. Considering the prominent effects on hepatic glucose and lipid metabolism and insulin sensitivity, O-GlcNAcylation may be significantly associated with the development and progression of liver-associated metabolic disorders. The exciting discovery of agents targeting O-GlcNAcylation and the HBP may provide a window of therapeutic opportunity. However, research on the treatment applications of O-GlcNAcylation has met with discrepancy and, thus, been slow to emerge. The most important reason is that O-GlcNAcylation is a highly dynamic process, and different methods used to manipulate O-GlcNAc signaling may lead to contradictory results. Tools and approaches quantifying protein O-GlcNAcylation are still under development. Moreover, altering the global level of O-GlcNAcylation is undesirable, which may influence O-GlcNAc on thousands of irrelevant proteins and counteract the actual effects of proteins of interest, thus increasing severe side effects or even introducing new issues. In addition, the level of O-GlcNAcylation as well as the dynamic cycling and the HBP pathway are critical in nutrient flux and many other physical processes, and global inhibition of OGT, OGA, or GAFT may cause disruptions and unwanted effects. Therefore, the O-GlcNAcylation of the protein of interest should be targeted, rather than a global alteration. Gene editing can introduce point mutations to abolish O-GlcNAc sites on a specific protein. Intriguingly, O-GlcNAc modification on different sites from the same protein may have completely opposite effects, reaffirming the importance of site-specific modulation [98,99]. Agents selectively blocking the interactions between specific proteins and OGT are also promising [127]. Taken together, these findings demonstrate that O-GlcNAcylation is critical in liver metabolism, and precise alteration of this modification is of therapeutic importance for metabolic disorders, including diabetes and NAFLD. Further work is required to develop new methods for accurately manipulating O-GlcNAcylation as well as new agents and techniques for realizing site-specific and protein-specific alterations, which will not only provide a novel approach for the treatment of liver-associated metabolic disorders, but also benefit the entire field of O-GlcNAcylation research.

## Figures and Tables

**Figure 1 ijms-24-02142-f001:**
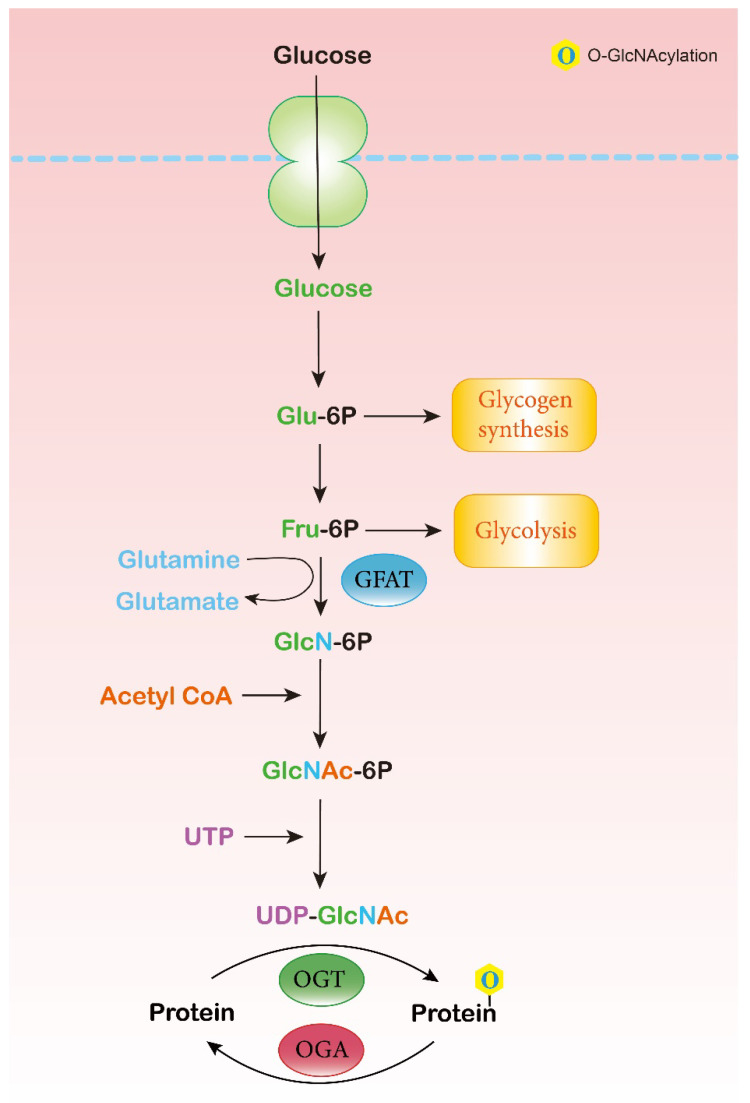
The hexosamine biosynthesis pathway (HBP) and O-GlcNAcylation. The HBP integrates glucose, glutamine, acetyl-CoA, aspartic acid, and UTP to synthesize the final product, UDP-GlcNAc, the glycosyl donor for O-GlcNAcylation. GFAT, glucosamine fructose-6-phosphate amidotransferase; GlcNAc, N-acetylglucosamine; UTP, uridine triphosphate; OGT, O-GlcNAc transferase; OGA, β-N-acetylglucosaminidase; O, O-GlcNAcylation.

**Figure 2 ijms-24-02142-f002:**
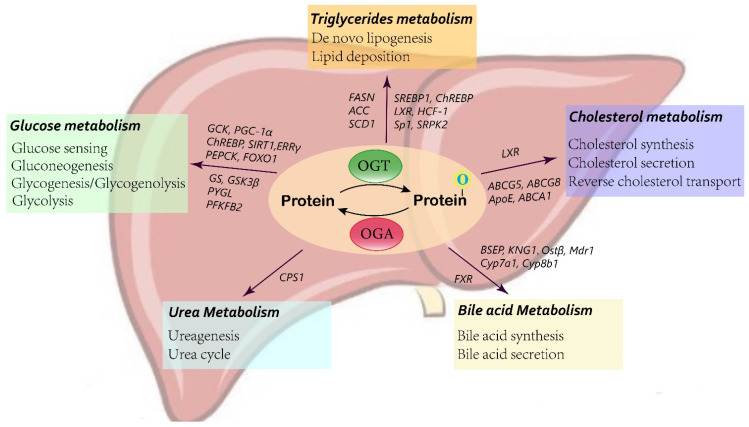
Schematic of O-GlcNAcylation in liver metabolism.

**Figure 3 ijms-24-02142-f003:**
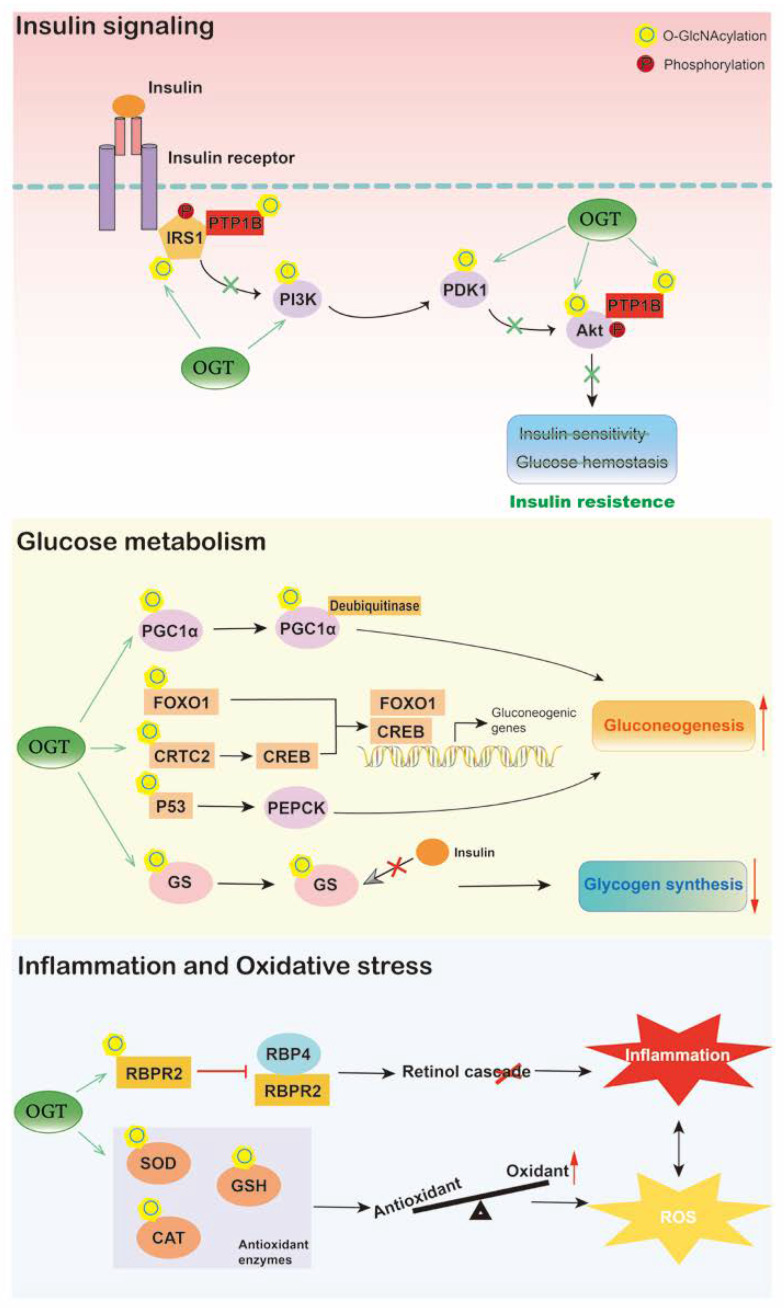
Hepatic O-GlcNAcylation plays an important role in diabetes. O-GlcNAcylation of proteins involved in insulin signaling contributes to liver insulin resistance. O-GlcNAc signaling also influences hepatic glucose metabolism, increased O-GlcNAcylation induces gluconeogenesis and inhibits glycogen synthesis, thus exaggerating hyperglycemia. Moreover, under high-glucose situation, O-GlcNAcylation promotes liver inflammation and oxidative stress by triggering retinol cascade disruption and inhibiting antioxidant activities, respectively, which contributes to diabetes development. O, O-GlcNAcylation; P, phosphorylation.

**Figure 4 ijms-24-02142-f004:**
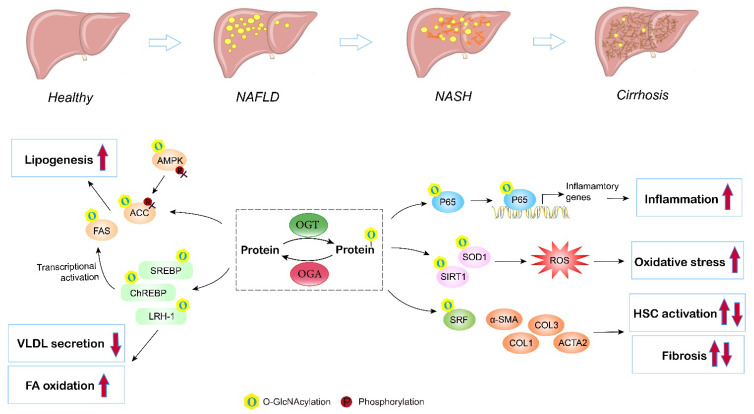
Role of O-GlcNAcylation in NAFLD/NASH development. Elevated O-GlcNAcylation induces de novo lipogenesis through direct regulation of the lipogenic target gene FAS and ACC, and by mediating the nuclear translocation of the master lipogenic transcription factors SREBP-1, ChREBP, and LXR. O-GlcNAcylation of LRH-1 promotes fatty acid oxidation and inhibits liver VLDL secretion. O-GlcNAc-modification of P65 induces translocation and activates NF-kB signaling, thus promoting liver inflammation and NASH. Additionally, O-GlcNAcylation enhances hepatic oxidative stress by triggering inhibiting activities of antioxidant enzymes (SIRT1 and SOD1). Moreover, studies have shown that O-GlcNAcylation influences hepatic stellate cell (HSC) activation, but the exact role (induce or inhibit) remains controversial. O, O-GlcNAcylation; P, phosphorylation.

## Data Availability

Not applicable.

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
