# Peer review of "Emerging Role of Protein O-GlcNAcylation in Liver Metabolism: Implications for Diabetes and NAFLD"

_ijms, 2023, doi:10.3390/ijms24032142_

Round 1

Reviewer 1 Report

This review, entitled -Protein O-GlcNAcylation: Emerging Role in Liver Metabolism and Associated Diseases- discusses the role of O-GlcNAcylation in liver metabolism and highlight the implications of O-GlcNAcylation in liver metabolic diseases: diabetes, non-alcoholic fatty liver disease and associated liver fibrosis, thus providing evidence for pathogenesis and potential therapeutic targets. This scientific collect is very interesting, however, minor problems, as indicated below, should be addressed before the document can be considered for publication in the this journal.

Minor problems:

English language and style are fine, minor spell check is required to ensure that an international audience can clearly understand your text. Moreover, I suggest to review the style of the manuscript according to the guidelines of the journal.

Glycosylation is involved in several diseases (doi: 10.1631/jzus.B1900150), including diabetic nephropathy. Thus, I suggest to add this reference (doi: 10.3389/fcell.2020.607080), in which the link between O-GlcNAcylation and kidney failure is demonstrated in the context of Diabetes.

In Figure 1 and 2, the authors should indicate the meaning of the vocal "O". Moreover, they should explain better the caption of the Figures.

Author Response

We thank the reviewers for the time and effort that they have put into reviewing the previous version of the manuscript. Their suggestions have enabled us to improve our work. We agree with these comments, and we have addressed these specific concerns point by point. We enabled the ‘Track changes’ feature in Microsoft Word and all changes made are easily identifiable in our paper.

  1. English language and style are fine, minor spell check is required to ensure that an international audience can clearly understand your text. Moreover, I suggest to review the style of the manuscript according to the guidelines of the journal.

-Answer: Thank you for your suggestions. This manuscript has been edited by professional editing service. The verification code is F0ED-EC0C-8F0E-BC12-7574. Please check the manuscript for details.

  1. Glycosylation is involved in several diseases (doi: 10.1631/jzus.B1900150), including diabetic nephropathy. Thus, I suggest to add this reference (doi: 10.3389/fcell.2020.607080), in which the link between O-GlcNAcylation and kidney failure is demonstrated in the context of Diabetes.

-Answer: Thank you for your advice. O-GlcNAcylation plays an important role in several diabetic complications (doi: 10.1111/jdi.13359). However, in this review we focused on the role of O-GlcNAcylation in liver metabolism and liver dysfunction-associated diabetes instead of diabetic complications. Therefore, this reference was added in the “Introduction” part, please check the manuscript for details.

  1. In Figure 1 and 2, the authors should indicate the meaning of the vocal "O". Moreover, they should explain better the caption of the Figures.

-Answer: Thank you for your suggestions. The meaning of "O" in figures has been explained and figure legends have been clarified, please check the manuscript for details.

Reviewer 2 Report

The topic of the review is of interest. The manuscript is focused on the function and regulation of O-GlcNAcylation in the liver metabolism and associated metabolic diseases. The manuscript was well-written and organized. However, although the title of the review manuscript is comprehensive, the scope of the review manuscript seems to narrow down since it is specifically focused on the role of O-GlcNAc in pathogenesis of diabetes and non-alcoholic fatty liver disease (NAFLD) among several liver diseases. In addition, there are many incorrect and missing references. Furthermore, English revision and minor linguistic correction are necessary. The file of detailed comments and suggestions are attached, please find it.

Author Response

We thank you for the time and effort that you have put into reviewing the previous version of the manuscript. Your suggestions have enabled us to improve our work. We agree with these comments and we have addressed these specific concerns point by point.

Major

Point 1: The title of the manuscript, “Protein O-GlcNAcylation: Emerging Role in Liver Metabolism and Associated Diseases”, would be seemed that it extensively covers the role of O-GlcNAcylation in several cellular signaling pathways correlated with liver metabolism and liver diseases including diabetes, non-alcoholic fatty liver disease (NAFLD), and liver fibrosis which leads to hepatocellular carcinoma (HCC). Up to date, it has been reported that O-GlcNAcylation is closely associated with various cancer types including HCC. However, the manuscript is only focused on the O-GlcNAcylation in diabetes and non-alcoholic fatty liver disease (NAFLD) in aspect of liver diseases. So, the reviewer would like to recommend the authors to alter the title of the manuscript more specifically.

-Answer: Thank you for your suggestions. The title has been altered as “Emerging Role of Protein O-GlcNAcylation in Liver Metabolism: Implications for Diabetes and NAFLD”.

Point 2: Figure 2. did not include additional information to describe O-GlcNAcylation of target proteins participated in liver metabolism. For example, although, in “3.1. Glucose Metabolism” of the manuscript, the authors described that O-GlcNAc of GCK, PGC-1α, FOXO1, ChREBP, and ERRγ mediates gluconeogenesis by altering the expression and activity of key gluconeogenic genes (PEPCK and G6P), ChREBP and ERRγ were not depicted in Figure 2.

-Answer: Thank you for your advice. All the target genes of O-GlcNAcylation involved in liver metabolism have been added in Figure 2. Please check the manuscript for details.

Minor

Point 1: Mark unified terminology, such as “post-translational”, “FAS”, and “PGC-1α”, throughout the manuscript, figures, and figure legends.

-Answer: Thank you for your suggestions. The terminology has been unified throughout the manuscript. Please check the manuscript for details.

Point 2: Revise incorrect information compared to the cited references in ‘Line 101-104’ ([25]), ‘Line 578-580’ ([103]). Point 3: Add appropriate references for description, ‘Line 36-47’, ‘Line 185-186’, ‘Line 193-197’, ‘Line 199-201’, ‘Line 284-287’, ‘Line 302-303’, ‘Line 399-402’, ‘Line 521-522’, ‘Line 560-562’, ‘Line 589-590’,‘Line 623-624’, and ‘Line 624-625’. Point 4: Correct the improper references, [9] (Line 70 and Line 84), [21] (Line 97), [27] (Line 171), [30] (Line 173-176), [37] (Line 190), [42] (Line 205), [85] (Line 385), and [15] (Line 596-601). Point 5: Cite more than 2 proper references for description, ‘Line 75’, ‘Line 199-201’, ‘Line 278-279’, and ‘Line 603-606’.Point 6: [8] and [27] are redundant.

-Answer: Thank you for your careful work. All the references as you mentioned have been revised. We deleted redundant references, replaced improper or incorrect references, and altered the original text to make sure that the description was consistent with the reference. Please check the manuscript for details (Highlighted).

Point 7: “X-Box Binding protein-s (XBP1s)” (Line 100-101), “Forkhead Box Other-1” (Line 262), “PCK” (Line 271), and “G6PD” (Line 497) should be changed in “X-Box Binding protein-1 (XBP-1)”, “Forkhead Box protein O1”, “PEPCK”, and “G6P”, respectively.

-Answer: These words and abbreviations have been corrected.

Point 8: In Figure 2, 3, and 4, “PCK”, “PIP1B”, and “LXR” should be changed in “PEPCK”, “PTP1B”, and “LRH-1”, respectively.

-Answer: The figures have been corrected.

Point 9: Line 244, “.3.1. Glucose Metabolism” should be changed in “3.1. Glucose Metabolism”.

-Answer: This part has been corrected.

Point 10: Try to reduce the overuse of plural nouns unless necessary.

Point 11: English correction are necessary throughout the manuscript.

-Answer: Thank you for your suggestions. This manuscript has been edited by professional editing service. The verification code is F0ED-EC0C-8F0E-BC12-7574.  Please check the manuscript for details.
